# DIRECTIONAL ENSEMBLE AGGREGATION FOR ACTOR-CRITICS

## ABSTRACT

Reliable $Q$-value estimation is central to off-policy reinforcement learning in continuous control. Standard actor-critic methods often address overestimation bias by aggregating ensembles of $Q$-values conservatively, for example by taking their minimum. While effective at reducing bias, these static rules discard useful information, cannot adapt to training dynamics, and generalize poorly across learning regimes. We propose Directional Ensemble Aggregation (DEA), a fully learnable aggregation method that replaces static aggregation with a dynamic mechanism capable of interpolating between conservative and explorative strategies as training progresses. DEA introduces two learnable directional parameters, one regulating critic conservatism and the other guiding actor exploration. Both are learned using disagreement-weighted Bellman errors, where updates depend only on the sign of each sample's error. This decoupled design allows DEA to adjust automatically to task-specific uncertainty, ensemble size, and update frequency in a data-driven manner. Empirically, DEA generalizes across MuJoCo and DeepMind Control Suite benchmarks in both interactive and sample-efficient learning regimes.

## 1 INTRODUCTION

Off-policy Reinforcement Learning (RL) has become a powerful framework for solving continuous control tasks, with actor-critics forming a central class (Mnih et al., 2013; 2015). These methods decompose training into two components: a *critic* ($Q$-function), which estimates the expected returns, and an *actor*, which uses these $Q$-value estimates to optimize its policy (Haarnoja et al., 2018a).

A central challenge in actor-critics is *overestimation bias* in $Q$-value estimation (Thrun & Schwartz, 1993), where value estimates drift upward because positive noise is repeatedly amplified through bootstrapping and the maximization operator (Fujimoto et al., 2018). Such bias can destabilize training and lead to suboptimal policies (Van Hasselt, 2010; Van Hasselt et al., 2016), especially in continuous control, where even small biases in $Q$-values may be exploited by the actor. Another challenge is *sample efficiency*. A common remedy is to increase the number of updates per environment interaction, known as the Update-To-Data (UTD) ratio (Chen et al., 2021). However, higher UTD ratios place greater demands on the accuracy of $Q$-values, as repeated updates per environment step can amplify estimation errors and further destabilize training (Nikishin et al., 2022).

To mitigate overestimation, many algorithms employ an ensemble of critics and aggregate their $Q$-value estimates conservatively, typically by taking the minimum across the ensemble (Fujimoto et al., 2018; Haarnoja et al., 2018a; Ciosek et al., 2019; Chen et al., 2021). While effective in reducing bias, these static aggregation rules have several drawbacks. First, they impose unnecessary restrictions that prevent smooth interpolation between conservative and explorative strategies, hindering fine-grained control over the exploration-exploitation trade-off. Second, they collapse ensemble diversity into a single value, discarding useful information. Third, they remain fixed throughout training, unable to adapt to task-specific demands or the evolving reliability of $Q$-value estimates. This rigidity also extends to policy optimization, where the actor is constrained to the same static aggregation rule regardless of the environment's exploration demands or the stage of training.

Furthermore, most existing algorithms are designed with a specific *learning regime*, defined by particular combinations of UTD ratios and ensemble sizes. For example, *interactive learning* typically employs low UTD ratios and small ensembles with frequent environment interactions (Fujimoto et al., 2018; Haarnoja et al., 2018a; Ciosek et al., 2019; Moskovitz et al., 2021). Conversely, *sample-efficient*

*learning* relies on higher UTD ratios and larger ensembles to minimize the number of interactions required to learn a task (Chen et al., 2021; Wu et al., 2022; Cetin & Celiktutan, 2023). Algorithms optimized for one learning regime often fail in the other: increasing the UTD ratio in methods designed for interactive learning can introduce instability and degrade performance (Nikishin et al., 2022), whereas methods optimized for sample-efficient learning tend to underperform in low-UTD scenarios due to insufficient utilization of available updates (Liang et al., 2022; Chen et al., 2021).

**Our approach.** We propose Directional Ensemble Aggregation (DEA), a fully learnable aggregation method for actor-critic algorithms that adapts dynamically to task demands and uncertainty levels. DEA overcomes the rigidity of existing methods that are tailored to specific learning regimes, enabling generalization across settings where static rules often fail. At its core, DEA replaces static rules with a data-driven mechanism that interpolates between conservative and explorative strategies as training progresses.

A key feature of DEA is its decoupled aggregation, introducing two learnable parameters: $\bar{\kappa}$, used to construct what guides critic learning, and $\kappa$, to construct what guides actor learning. Separate parameters are essential because, especially under high uncertainty, critics often benefit from more conservative estimates to reduce overestimation bias, while actors can exploit more optimistic estimates when uncertainty is low to encourage exploration.

These parameters are learned directly from data through some weighted Bellman objectives, with updates depending only on the *sign* of each sample's error. The sign reliably indicates whether an estimate overshoots or undershoots its target, while ignoring noisy magnitudes that could otherwise cause disproportionate updates. By relying on this directional signal, DEA avoids drastic parameter swings and achieves stable, data-driven adaptation of aggregation. This reliance on error direction motivates the name *directional* ensemble aggregation.

Across a broad set of continuous control benchmarks in MuJoCo and the DeepMind Control Suite, DEA consistently demonstrates strong generalization by maintaining reliable learning dynamics and outperforming static ensemble aggregation methods in both interactive and sample-efficient settings.

## 2 PRELIMINARY

We denote by $\mathcal{P}(\Omega)$ the set of all probability distributions over a space $\Omega$, and by $\mathcal{B}(\Omega)$ the set of bounded real-valued functions on $\Omega$. For $N \in \mathbb{N}$, we write $[N]$ for $\{1, \ldots, N\}$.

**Markov Decision Processes (MDPs).** We consider an infinite-horizon MDP defined by the tuple $\mathcal{M} = \langle \mathcal{S}, \mathcal{A}, p, p_0, r, \gamma \rangle$ (Puterman, 2014), where $\mathcal{S}$ and $\mathcal{A}$ are continuous state and action spaces. The transition dynamics are governed by an unknown probability density $p(s'|s, a)$ over next states $s' \in \mathcal{S}$ given a current state-action pair $(s, a) \in \mathcal{S} \times \mathcal{A}$. The initial state is drawn from a distribution $p_0 \in \mathcal{P}(\mathcal{S})$, and rewards are given by a bounded function $r : \mathcal{S} \times \mathcal{A} \to [0, B_r]$, with $B_r > 0$. The discount factor $\gamma \in (0, 1]$ controls the importance of future rewards.

**Policies.** Let $\Pi = \{\pi : \mathcal{S} \to \mathcal{P}(\mathcal{A})\}$ denote the set of stochastic policies. Under a policy $\pi \in \Pi$, the agent interacts with the MDP iteratively: at each time step $t \in \mathbb{N}$, the agent observes a state $s_t \in \mathcal{S}$, samples an action $a_t \sim \pi(\cdot|s_t)$, receives a reward $r(s_t, a_t)$, and transitions to a next state $s_{t+1} \sim p(\cdot|s_t, a_t)$. For convenience, we define the one-step policy-induced transition distribution as $p^\pi(s', a'|s, a) = p(s'|s, a)\pi(a'|s')$.

**Maximum entropy RL.** The objective is to find a policy $\pi$ that maximizes the expected discounted return, $J(\pi) = \mathbb{E}_\pi[\sum_{t=0}^\infty \gamma^t r(s_t, a_t)]$ with $s_0 \sim p_0$ (Sutton & Barto, 2018; Bertsekas & Tsitsiklis, 1996). To encourage exploration, we consider the maximum entropy RL framework (Ziebart, 2010; Haarnoja et al., 2017; 2018a), where the agent aims to maximize both the expected return and the entropy of the policy. This is formalized by the objective: $J_\alpha(\pi) = \mathbb{E}_\pi \left[ \sum_{t=0}^\infty \gamma^t \left( r(s_t, a_t) + \alpha \mathcal{H}(\pi(\cdot|s_t))) \right) \right]$, where $\alpha > 0$ controls the trade-off between reward and entropy, and the entropy term is defined as $\mathcal{H}(\pi(\cdot|s)) = -\mathbb{E}_{a \sim \pi(\cdot|s)}[\log \pi(a|s)]$. Haarnoja et al. (2018b) proposed automatic entropy tuning by adjusting $\alpha$ to minimize the objective $J(\alpha) = \mathbb{E}_{a \sim \pi(\cdot|s)} [\log(\alpha) \cdot (-\log \pi(a|s) - \mathcal{H}_{\text{target}})]$ during each policy update. This objective increases $\alpha$ when the current policy entropy is below the target $\mathcal{H}_{\text{target}}$, and to decrease

when it is above. In practice, the target entropy $\mathcal{H}_{\text{target}}$ is typically set heuristically, proportional to $-\dim(\mathcal{A})$ or $-\dim(\mathcal{A})/2$ (Haarnoja et al., 2018b; Chen et al., 2021). The state-action value function $Q^\pi : \mathcal{S} \times \mathcal{A} \to \mathbb{R}$ under policy $\pi$ satisfies $Q^\pi(s,a) = J_\alpha(\pi)$ starting from $s_0 = s$ and $a_0 = a$ (Watkins & Dayan, 1992; Haarnoja et al., 2017).

## 3 SOFT ACTOR-CRITIC METHODS

A widely used framework for maximum entropy RL is the class of soft actor-critic methods, where the agent jointly learns a $Q$-value function (the critic) and a policy (the actor). The active *critic* $Q$ estimates the state-action value function $Q^\pi$ (following policy $\pi$) by minimizing the Bellman error (Silver et al., 2014; Bertsekas & Tsitsiklis, 1996; Haarnoja et al., 2018a):

$$\arg \min_Q \mathbb{E}_{(s,a,r,s')\sim\mathcal{D}}[(Q(s,a) - y(s,a,r,s'))^2], \qquad (1)$$

where the samples $(s,a,r,s')$ are drawn from a replay buffer $\mathcal{D}$, and the critic's *target value* $y(s,a,r,s')$ is defined as:

$$y(s,a,r,s') = r(s,a) + \gamma[\bar{Q}(s',a') - \alpha \log \pi(a'|s')], \quad a' \sim \pi(\cdot|s'), \qquad (2)$$

with $\bar{Q}$ denoting the *target critic* that may differ from the active critic $Q$ (Mnih et al., 2015). The *actor* is trained to find a policy $\pi \in \Pi$ that maximizes the expected entropy-regularized value:

$$\arg \max_{\pi \in \Pi} \mathbb{E}_{a\sim\pi(\cdot|s)}[\tilde{Q}(s,a) - \alpha \log \pi(a|s)], \qquad (3)$$

where $\tilde{Q}$ is a separate *actor-update critic* used solely for training the actor, and may differ from both the active critic $Q$ and the target critic $\bar{Q}$.

Note that $\bar{Q}$ and $\tilde{Q}$ are *references*: $\bar{Q}$ guides critic learning (in (2)), while $\tilde{Q}$ guides actor learning (in (3)). Although they are not the critic or actor themselves, their construction strongly affects stability, bias, and training success. Thus, when we later describe the critic or actor as *conservative* or *optimistic/explorative*, we refer to how these references are constructed. In the idealized setting with exact Bellman policy evaluation, all three value estimators, $Q$, $\bar{Q}$, and $\tilde{Q}$, would match the true value function $Q^\pi$ (Sutton & Barto, 2018). However, different actor-critic methods, such as SAC (Haarnoja et al., 2018a) and REDQ (Chen et al., 2021), differ primarily in how they construct $\bar{Q}$ and $\tilde{Q}$.

In Section 4, we review how existing soft actor-critic methods instantiate these components and highlight their limitations. Next, in Section 5, we introduce our proposed DEA approach, which generalizes these constructions through a fully learnable and adaptive ensemble aggregation strategy.

## 4 ENSEMBLE AGGREGATION IN SOFT ACTOR-CRITIC METHODS

Ensembles of $Q$-functions have become standard in modern actor-critic algorithms for improving stability and mitigating overestimation bias. Most methods aggregate the outputs of these critics using conservative rules (e.g. by taking the minimum across the ensemble) and pair this with delayed copies as target networks to stabilize updates (Lillicrap et al., 2016; Mnih et al., 2015; Fujimoto et al., 2018). Formally, for each $i \in [N]$, let $Q_i : \mathcal{S} \times \mathcal{A} \to \mathbb{R}$ be the (active) critic and $\bar{Q}_i : \mathcal{S} \times \mathcal{A} \to \mathbb{R}$ its corresponding (delayed) target critic.

The minimum strategy, introduced by Fujimoto et al. (2018) and adopted by SAC (Haarnoja et al., 2018a), aggregates the ensemble by selecting the lowest $Q$-value estimate. SAC employs this rule for both the critic target, $\bar{Q}$, used in (2), and the actor-update value, $\tilde{Q}$, used in (3);

SAC: $$\bar{Q}(s,a) = \min_{i\in[N]} \bar{Q}_i(s,a) \quad \text{and} \quad \tilde{Q}(s,a) = \min_{i\in[N]} Q_i(s,a).$$

SAC works well for small ensembles (e.g., $N = 2$), but becomes overly conservative as $N$ grows, often leading to underestimation and overly cautious policies (Lan et al., 2020; Kuznetsov et al., 2020). Instead, Chen et al. (2021) proposed REDQ, which maintains a larger ensemble of $N = 10$ critics and applies different aggregation strategies for critic and actor. At each critic update, REDQ

samples a random subset $S \subset [N]$ of size $|S| = 2$ and uses the minimum over this subset as the target critic, and for actor policy updates it uses the average over the full $N$-ensemble;

$$\text{REDQ:} \qquad \bar{Q}(s,a) = \min_{i \in S} \bar{Q}_i(s,a) \quad \text{and} \quad \tilde{Q}(s,a) = \frac{1}{N} \sum_{i=1}^{N} Q_i(s,a).$$

This decoupled aggregation balances critic conservatism (via a minimum of a random subset) with actor expressiveness (via a full ensemble average), providing improved training stability and greater sample efficiency compared to SAC, particularly in high UTD ratio regimes.

**Limitations.** SAC and REDQ share structural limitations. Both rely on static aggregation rules for $\bar{Q}$ and $\tilde{Q}$, which do not generalize well across learning regimes. SAC's use of a fixed minimum becomes increasingly conservative with larger ensembles, often leading to persistent underestimation (Nikishin et al., 2022), while REDQ's actor-side averaging can become unstable when the ensemble is small.

## 5    DIRECTIONAL ENSEMBLE AGGREGATION IN SOFT ACTOR-CRITIC METHODS

Directional Ensemble Aggregation (DEA) introduces a fully learnable framework for ensemble-based value estimation in actor-critic methods. It generalizes static aggregation strategie, such as those used in SAC and REDQ, by learning reference values in a data-driven manner that adapts to uncertainty and training dynamics. The adaptation is controlled by two scalar parameters: $\bar{\kappa}$, which controls critic learning stability, and $\kappa$, which modulates actor exploration. Both are learned online based on the ensemble's internal disagreement. Table 1 summarizes how DEA compares to existing methods.

Table 1: Aggregation strategies for constructing the target critic ($\bar{Q}$) and the actor-update critic ($\tilde{Q}$). $\bar{\delta}$ and $\delta$ denote ensemble disagreement among $\{\bar{Q}_i\}$ and $\{Q_i\}$, respectively. $\bar{\kappa}$ and $\kappa$ are DEA's learnable parameters that determine how conservative or optimistic $\bar{Q}$ and $\tilde{Q}$ should be.

| Method | Target critic $\bar{Q}$ | Actor-update critic $\tilde{Q}$ |
|---|---|---|
| SAC | $\min_{i \in [N]} \bar{Q}_i$ | $\min_{i \in [N]} Q_i$ |
| REDQ | $\min_{i \in S} \bar{Q}_i$ | $\frac{1}{N} \sum_{i=1}^{N} Q_i$ |
| DEA | $\frac{1}{N} \sum_{i=1}^{N} \bar{Q}_i + \bar{\kappa} \cdot \bar{\delta}$ | $\frac{1}{N} \sum_{i=1}^{N} Q_i + \kappa \cdot \delta$ |

DEA integrates seamlessly into the soft actor-critic framework (Section 3); its full update cycle is outlined in Algorithm 1, with the design rationale discussed below.

---

**Algorithm 1** Directional Ensemble Aggregation (DEA)

1: **Initialize:** replay buffer $\mathcal{D} = \emptyset$; critic networks $Q_{\theta_1}, \ldots, Q_{\theta_N}$ and actor network $\pi_\phi$ with random parameters $\{\theta_i\}_{i=1}^{N}$ and $\phi$; target critic networks $\bar{Q}_{\bar{\theta}_1}, \ldots, \bar{Q}_{\bar{\theta}_N}$ with $\bar{\theta}_i \leftarrow \theta_i$ for $i = 1, \ldots, N$
2: **for** each environment interaction **do**
3:     take action $a_t \sim \pi_\phi(\cdot|s_t)$, observe reward $r_t \triangleq r(s_t, a_t)$, transition to new state $s_{t+1} \sim p(\cdot|s_t, a_t)$, and add $(s_t, a_t, r_t, s_{t+1})$ to replay buffer $\mathcal{D}$
4:     **for** each update-to-data ratio **do**
5:         sample mini-batch $B = \{(s, a, r, s', a') : (s, a, r, s') \sim \mathcal{D}, a' \sim \pi_\phi(\cdot|s')\}$
6:         $\theta_i \leftarrow \theta_i - \eta_\theta \nabla_\theta \{\frac{1}{|B|} \sum_B (Q_{\theta_i}(s,a) - y_{\bar{\kappa}}(s,a,r,s'))^2\}, \ \forall i \in [N]$       ▷ *critic*
7:         $\bar{\theta}_i \leftarrow \tau\theta_i + (1-\tau)\bar{\theta}_i, \ \forall i \in [N]$       ▷ *target critic*
8:     $\bar{\kappa} \leftarrow \bar{\kappa} - \eta_{\bar{\kappa}} \nabla_{\bar{\kappa}} \{\frac{1}{|B|} \sum_B |\tilde{Q}_\kappa(s,a) - y_{\bar{\kappa}}(s,a,r,s')|/\bar{\delta}(s',a')\}$     ▷ *DEA: target critic*
9:     $\kappa \leftarrow \kappa - \eta_\kappa \nabla_\kappa \{\frac{1}{|B|} \sum_B |\tilde{Q}_\kappa(s,a) - y_{\bar{\kappa}}(s,a,r,s')|/\delta(s,a)\}$     ▷ *DEA: actor*
10:     $\phi \leftarrow \phi + \eta_\phi \nabla_\phi \{\frac{1}{|B|} \sum_B (\tilde{Q}_\kappa(s, a_\phi(s)) - \alpha \log \pi_\phi(a_\phi(s)|s))\}, \ a_\phi(s) \sim \pi_\phi(\cdot|s)$  ▷ *policy*
11:     $\alpha \leftarrow \alpha + \eta_\alpha \nabla_\alpha \{\frac{1}{|B|} \sum_B (\log(\alpha)(-\log \pi_\phi(a|s) - \mathcal{H}_{\text{target}}))\}$     ▷ *entropy*

---

**Ensemble disagreement.** To quantify uncertainty within the ensemble, DEA uses a measure of ensemble disagreement defined by the average pairwise deviation of $Q$-value estimates. Specifically, for a state-action pair $(s, a)$:

$$\delta(\{Q_i(s,a)\}_{i=1}^{N}) = \frac{1}{\binom{N}{2}} \sum_{i>j} |Q_i(s,a) - Q_j(s,a)|, \quad \text{for } i, j \in [N].$$

This metric is non-parametric, easy to compute, and does not rely on distributional critics (Bellemare et al., 2023). Related notions of ensemble diversity are discussed in more detail in Section 8. For notational clarity, we distinguish disagreement among target critics and active critics:

$$\bar{\delta}(s,a) \triangleq \delta(\{\bar{Q}_i(s,a)\}_{i=1}^N) \quad \text{and} \quad \delta(s,a) \triangleq \delta(\{Q_i(s,a)\}_{i=1}^N).$$

**Soft actor-critic learning with DEA.** DEA is built on the general soft actor-critic framework (Section 3), but can be easily generalized to other algorithmic families. We chose SAC as a use case, as it is the only commonly adopted algorithm with established variants tailored to the learning regimes considered in this work. It replaces fixed aggregation with learnable, uncertainty-aware versions of the target critic $\bar{Q}$ and the actor-update critic $\tilde{Q}$:

$$\text{DEA:} \quad \bar{Q}_{\bar{\kappa}}(s,a) = \frac{1}{N}\sum_{i=1}^N \bar{Q}_i(s,a) + \bar{\kappa}\cdot\bar{\delta}(s,a) \quad \text{and} \quad \tilde{Q}_{\kappa}(s,a) = \frac{1}{N}\sum_{i=1}^N Q_i(s,a) + \kappa\cdot\delta(s,a).$$

Instead of (2), the critic is trained using the modified *target value* using $\bar{Q}_{\bar{\kappa}}$:

$$y_{\bar{\kappa}}(s,a,r,s') = r(s,a) + \gamma[\bar{Q}_{\bar{\kappa}}(s',a') - \alpha\log\pi(a'|s')], \quad a' \sim \pi(\cdot|s'). \tag{4}$$

As in previous work (see e.g., Section 3), DEA ensures stability through delayed target critics to prevent rapid shift. Also, all critics share the same ensemble-based target value (4), promoting consistency across the ensemble and avoiding instability caused by divergent critic objectives.

For policy optimization, the actor trained using the modified *actor-update critic* $\tilde{Q}_{\kappa}$ estimate:

$$\arg\max_{\pi\in\Pi}\mathbb{E}_{a\sim\pi(\cdot|s)}[\tilde{Q}_{\kappa}(s,a) - \alpha\log\pi(a|s)]. \tag{5}$$

**Learning the directional aggregation parameters.** DEA update $\bar{\kappa}$ and $\kappa$ through a two-stage learning scheme that aligns critic and actor references while adapting to uncertainty. The critic-side parameter $\bar{\kappa}$ is optimized to stabilize target values by minimizing the ensemble disagreement-weighted Bellman error:

$$\arg\min_{\bar{\kappa}}\mathbb{E}_{(s,a,r,s')\sim\mathcal{D},a'\sim\pi(\cdot|s')}[|\tilde{Q}_{\kappa}(s,a) - y_{\bar{\kappa}}(s,a,r,s')|/\bar{\delta}(s',a')]. \tag{6}$$

Subsequently, the actor-side parameter $\kappa$ is updated to track the learned critic target while being regularized by active-critic disagreement:

$$\arg\min_{\kappa}\mathbb{E}_{(s,a,r,s')\sim\mathcal{D},a'\sim\pi(\cdot|s')}[|\tilde{Q}_{\kappa}(s,a) - y_{\bar{\kappa}}(s,a,r,s')|/\delta(s,a)]. \tag{7}$$

These objectives use disagreement-weighted absolute errors, producing *sign*-based gradients. Specifically, the gradient of (6) with respect to $\bar{\kappa}$ is $-\gamma\mathbb{E}[\text{sign}(\tilde{Q}_{\kappa}(s,a) - y_{\bar{\kappa}}(s,a,r,s'))]$, and the gradient of (7) with respect to $\kappa$ is $\mathbb{E}[\text{sign}(\tilde{Q}_{\kappa}(s,a) - y_{\bar{\kappa}}(s,a,r,s'))]$.

The key idea is that while error magnitudes are noisy, their *sign* reliably indicates whether estimates overshoot or undershoot targets. By using only this directional signal, DEA avoids domination by high-variance samples and ensures each transition contributes similarly. As a result, $\bar{\kappa}$ and $\kappa$ change gradually rather than erratically, improving stability. This reliance on error *direction* but no magnitude motivates the name: *directional* ensemble aggregation.

**Ensemble disagreement and its effect on conservatism and exploration.** Early in training, limited data and unrefined critics lead to high disagreement across the ensemble. As learning progresses and the critics become more aligned, disagreement decreases; especially relative to the growing scale of the $Q$-values (e.g., see Figures 10 to 13 in Appendix C). This evolving disagreement regulates the balance between conservatism and exploration in DEA and helps mitigate primacy bias (Nikishin et al., 2022) by reducing the influence of noisy early estimates.

When updating the critic parameters $\theta$, disagreement enters through the reference target $y_{\bar{\kappa}}$: if $\bar{\delta}(s',a')$ is large, a positive $\bar{\kappa}$ inflates $y_{\bar{\kappa}}$ and risks overestimation, so smaller (often negative) values yield more stable targets. When updating the actor parameters $\phi$, disagreement enters via the actor reference $\tilde{Q}_{\kappa}$: when $\delta(s,a)$ is large, large $\kappa$ would over-emphasize noisy estimates; only as disagreement falls does a larger $\kappa$ become reasonable, supporting more optimistic updates.

When updating the $\kappa$ parameters themselves, since the entropy term $-\alpha \log \pi(a'|s')$ is positive, $y_{\bar{\kappa}}$ often exceeds $\tilde{Q}_\kappa$, making $\tilde{Q}_\kappa - y_{\bar{\kappa}}$ negative on average. This drives $\bar{\kappa}$ downward (more conservative critic targets) while pushing $\kappa$ upward (more optimistic actor updates). Together, these dynamics ensure DEA transitions from cautious critic guidance under high uncertainty to more exploratory actor updates as training progresses.

The joint evolution of $\bar{\kappa}$ and $\kappa$ is analyzed in the next section and illustrated in Figure 1, with additional results across regimes and environments in Figures 6 to 9 in Appendix C.

## 6 EXPERIMENTS

**Learning regimes.** The goal of our experiments is to evaluate DEA across learning settings. These regimes are defined by the UTD ratio, which specifies how many gradient updates are performed per environment interaction. We consider two regimes, interactive and sample-efficient, ranging from low to high update intensity. To align with each setting, we scale the ensemble size proportionally to the UTD ratio; smaller ensembles minimize compute in interactive settings, while larger ensembles promote stability when updates are frequent. Table 2 summarizes the learning regimes. SAC is typically used for the interactive, while REDQ is designed for sample-efficient one. DEA is evaluated in both to assess its ability to generalize.

Table 2: Learning regimes.

| Learning regime | Ensemble size | Environment interactions | UTD ratio |
|---|---|---|---|
| Interactive | 2 | 1.000.000 | 1 |
| Sample-efficient | 10 | 300.000 | 20 |

**Evaluation metrics.** We evaluate performance using three metrics: Final return measures the average return at the end of training across evaluation repetitions; InterQuartile Mean (IQM) of the final evaluation-time return provides a robust average across seeds by excluding outliers (Agarwal et al., 2021); Area Under the Learning Curve (AULC) captures the cumulative reward over the course of training, reflecting both speed and stability of learning. For each metric and environment, we assign a rank to each method, where a lower rank indicates better performance (i.e., rank 1 is best, rank 2 is second-best, and so on). The average rank is then computed for each metric.

**Experimental setup.** We evaluate DEA on continuous control tasks from the MuJoCo physics simulator (Todorov et al., 2012; Brockman et al., 2016; Towers et al., 2024, version v5) and the DeepMind Control Suite (DMC) (Tassa et al., 2018), under the learning regimes defined in Table 2. These benchmarks were chosen because the limitations of static aggregation and the advantages of adaptive strategies are already well pronounced in such settings, and qualitatively different behavior is not expected in other continuous control domains. Each experiment is repeated across ten seeds. All methods use automatic entropy temperature tuning of $\alpha$, following Haarnoja et al. (2018b). Further details on network architecture, training settings, and hyperparameters are provided in Appendix B. Code will be released publicly upon publication to ensure reproducibility.

**Learning trajectories of directional parameters $\bar{\kappa}$ and $\kappa$.** To better understand how DEA modulates aggregation during training, Figure 1 visualizes the trajectories of $\bar{\kappa}$ and $\kappa$ under the two learning regimes (Table 2). We initialize the aggregation parameters as $\bar{\kappa} = -0.8$ and $\kappa = 0.0$; an ablation of initialization sensitivity is provided in Section 7. To ensure stable optimization in an unconstrained parameter space, we apply a tangent transformation to map both parameters into the open interval from minus one to one. The first row of the figure corresponds to the interactive regime, and the second row to the sample-efficient regime. The trends align with the behavior anticipated from our analysis in Section 5. In particular, as training progresses, $\bar{\kappa}$ typically remains negative, anchoring conservative critic estimates, while $\kappa$ tends to be positive and further increases when learning proceeds, to support explorative actor behavior. This dynamic interplay reflects DEA's ability to adaptively balance exploration and conservatism based on the ensemble disagreement and training context. Trajectories across all tasks, seeds, and learning regimes can be found in Appendix C.

**Generalization across learning regimes.** Average performance across both learning regimes is summarized in Table 3; detailed results for each individual regime are provided in Appendix C. Overall, DEA outperforms the best-performing baseline across tasks and metrics. Notably, it achieves

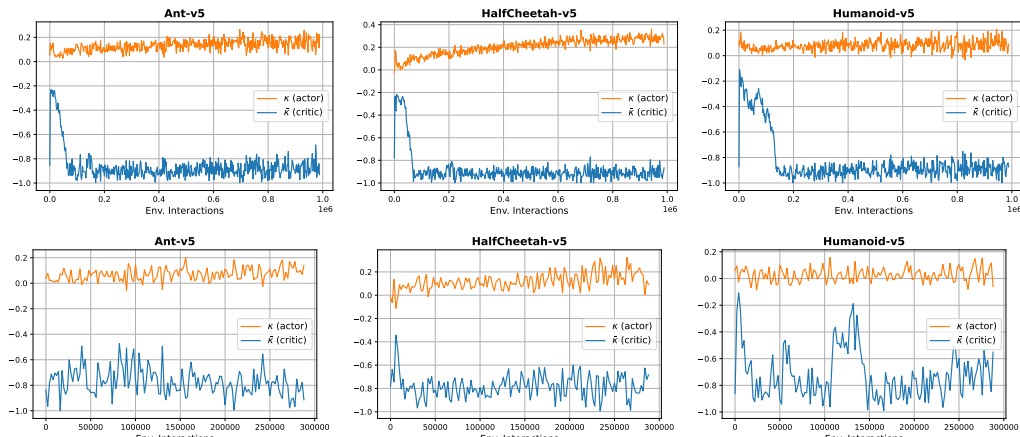

Figure 1: Trajectories of the directional aggregation parameters $\bar{\kappa}$ (critic) and $\kappa$ (actor) on MuJoCo. The top row shows results for the interactive regime, and the bottom row for the sample-efficient regime (Table 2). Trajectories of all tasks, seeds, and learning regimes can be found in Appendix C.

Table 3: Performance on MuJoCo and DMC environments. Metrics are average final return, InterQuantile Mean (IQM) of the final return, and Area Under the Learning Curve (AULC), averaged over learning regimes, evaluation repetitions, and ten seeds. Average rank is computed per metric across environments. ↑: higher is better, ↓: lower is better. Best algorithm per metric is **bold**.

| | Environment | Final Return (↑) | | | IQM (↑) | | | AULC (↑) | | |
| | | DEA | REDQ | SAC | DEA | REDQ | SAC | DEA | REDQ | SAC |
|---|---|---|---|---|---|---|---|---|---|---|
| MuJoCo | Ant-v5 | **4920** | 4278 | 2199 | **5226** | 4712 | 2583 | **2829** | 2249 | 1556 |
| | HalfCheetah-v5 | **10158** | 10065 | 8267 | **10334** | 10039 | 8250 | **7745** | 7682 | 6600 |
| | Hopper-v5 | **3376** | 2756 | 2526 | **3543** | 2835 | 2760 | **2780** | 2432 | 1891 |
| | Humanoid-v5 | **5073** | 4682 | 4832 | 5331 | **5338** | 5311 | **3315** | 2864 | 2857 |
| | Walker2d-v5 | 4425 | **4440** | 2989 | **4663** | 4564 | 2716 | **3111** | 3037 | 1963 |
| | MuJoCo Avg. Rank (↓) | **1.2** | 2.0 | 2.8 | **1.2** | 1.8 | 3.0 | **1.0** | 2.0 | 3.0 |
| DMC | Cheetah-run | 813 | **833** | 690 | 812 | **855** | 707 | 619 | **650** | 525 |
| | Hopper-hop | **175** | 174 | 58 | **157** | 148 | 40 | **112** | 105 | 32 |
| | Hopper-stand | **785** | 620 | 379 | **930** | 525 | 448 | **579** | 490 | 216 |
| | Humanoid-run | **152** | 149 | 80 | 154 | **155** | 80 | **92** | 83 | 39 |
| | Humanoid-stand | **658** | 634 | 348 | **676** | 654 | 313 | **344** | 327 | 170 |
| | Humanoid-walk | **498** | 487 | 283 | **507** | 506 | 283 | **282** | 267 | 149 |
| | Quadruped-run | **823** | 803 | 531 | 844 | **869** | 454 | **621** | 597 | 413 |
| | Quadruped-walk | **925** | 917 | 566 | **946** | 939 | 527 | **754** | 701 | 472 |
| | Walker-run | **731** | 705 | 503 | **753** | 715 | 499 | **597** | 568 | 395 |
| | DMC Avg. Rank (↓) | **1.11** | 1.89 | 3.00 | **1.33** | 1.67 | 3.00 | **1.11** | 1.89 | 3.00 |
| | Avg. Rank (↓) | **1.14** | 1.93 | 2.93 | **1.29** | 1.71 | 3.00 | **1.07** | 1.93 | 3.00 |

the highest average rank in all three metrics: final return, IQM, and AULC, with a particularly strong lead in AULC, indicating more reliable and efficient learning over time. Unlike existing methods, which are typically optimized for a fixed learning regime, DEA adapts its aggregation strategy during training, based on ensemble disagreement and training context, enabling it to generalize across learning regimes.

In particular, SAC is designed for low-UTD settings with small ensembles and often becomes overly conservative under higher UTD ratios. REDQ, by contrast, performs well in sample-efficient settings where its high UTD ratio and large ensemble size support rapid learning. Yet, REDQ tends to be unstable or less effective in low-update regimes, where its fixed update strategy becomes less reliable. DEA avoids these limitations by dynamically adjusting its behavior to the demands of the training context, rather than relying on static aggregation rules.

Learning curves and trajectories of the directional aggregation parameters ($\bar{\kappa}$ and $\kappa$) across all tasks and learning regimes are provided in Appendix C.

# 7 ABLATIONS

Our experiments already spans two distinct learning regimes (interactive and sample-efficient) which naturally serve as ablations over ensemble size and UTD ratio. As shown in Section 6, DEA remains effective across these varying configurations.

**Fixed aggregation and degenerate cases.** A natural ablation is to fix $\bar{\kappa}$ and $\kappa$ to constants. Doing so effectively recovers existing baselines: for example, setting both to enforce a static minimum corresponds to the conservative update used in SAC, while using a fixed mean aligns with REDQ. These variants remove DEA's adaptivity and result in behavior already covered by prior methods. Fixing either $\bar{\kappa}$ or $\kappa$ individually also degrades performance. As described in Section 5, these parameters act as directional anchors: $\bar{\kappa}$ controls critic-side conservatism, while $\kappa$ guides actor-side aggregation. Learning both jointly is crucial to allow DEA to interpolate between exploration and caution based on ensemble disagreement. Without this flexibility, the critic and actor can become misaligned, undermining the value of directional ensemble learning. In such cases, DEA loses its ability to adapt to uncertainty and learning dynamics, and its performance suffers accordingly.

**Sensitivity to initializations.** We study the sensitivity of DEA to different initializations of the critic-side aggregation parameter $\bar{\kappa}$, while keeping the actor-side initialization of $\kappa$ fixed at zero. This isolates the effect of $\bar{\kappa}$ and avoids interactions between the two directional parameters. Initializing $\kappa$ at zero provides a neutral initialization that does not bias the actor toward conservatism or optimism at the start of training. Results are shown in Figures 14 and 15 (Appendix D) for the five MuJoCo environments. As expected, the impact of initialization varies across tasks and learning regimes. While these results highlight the importance of choosing a good initialization, especially in higher UTD settings, DEA remains stable across all tested configurations, thanks to its robustness to the unforeseen effects of numerical perturbations. This robustness stems from its directional update rule, which only preserves the essential sign-information. None of the initializations lead to divergence or collapse, and many configurations outperform both SAC and REDQ.

# 8 RELATED WORK

Ensemble methods in off-policy RL have been widely studied for their ability to stabilize training, estimate uncertainty, and facilitate exploration. Below, we review prior work through three lenses most relevant to our work: (i) reducing overestimation bias, (ii) enabling efficient exploration, and (iii) improving sample efficiency across varying UTD regimes. These objectives often overlap, and many methods address multiple goals simultaneously. Below, we highlight representative approaches within each category. A more detailed discussion is provided in Appendix A.

**Reducing overestimation bias with ensembles.** Overestimation bias in $Q$-value estimates can destabilize training and degrade policy performance. To address this issue, various methods have been proposed for both discrete and continuous control. In discrete-action settings, overestimation is mitigated by aggregating multiple $Q$-value estimates in various different manners (Van Hasselt, 2010; Van Hasselt et al., 2016; Anschel et al., 2017; Lan et al., 2020). However, these techniques do not extend naturally to continuous control. In continuous settings, TD3 (Fujimoto et al., 2018) proposed taking the minimum over two $Q$-networks to reduce bias, a strategy adopted and extended by SAC (Haarnoja et al., 2018a) and others (Chen et al., 2021; Ciosek et al., 2019). These methods stabilize training by conservatively anchoring the target through fixed ensemble aggregation rules. DEA, on the other hand, avoids such hard-coded ensemble rules and allows adjusting ensembles dynamics based on training dynamics.

Recently, regularized value estimation methods have sought to adaptively reduce bias. GPL (Cetin & Celiktutan, 2023), for example, uses a distributional critic and dual TD-learning with regularization. However, GPL does not straightforwardly generalize outside its narrow learning regime, struggling in interactive (low-UTD) settings and requiring large ensembles and high UTD ratios for stability (Cetin

& Celiktutan, 2023, Figures 13 and 15). Its sample efficiency depends on pre-tuned, task-specific hyperparameters, particularly heuristic entropy targets, and a fixed optimism schedule (Cetin & Celiktutan, 2023, Tables 2 and 3). In contrast, DEA avoids such rigidity by using scalar $Q$-values and learning aggregation weights directly from data, adapting flexibly across learning regimes without distributional modeling or handcrafted schedules.

**Efficient exploration.**  While conservative value estimates provide stability, they can suppress exploration and slow learning. Prior approaches for discrete control encourage exploration by leveraging ensemble disagreement (Osband et al., 2016; Chen et al., 2017). In continuous control, OAC (Ciosek et al., 2019) leverages upper confidence bounds to guide exploration, while TOP (Moskovitz et al., 2021) frames the trade-off between conservatism and exploration as a multi-armed bandit problem, switching between predefined aggregation strategies. DAC (Nauman & Cygan, 2025) takes a different approach by maintaining two actors; a pessimistic one for conservative updates and an optimistic one for exploration. DEA differs from these methods by guiding a single actor through a fully learnable aggregation scheme. Rather than relying on handcrafted decision rules, confidence bounds, or architectural complexity, DEA integrates ensemble disagreement directly into the actor update and adjusts exploration adaptively during training.

**Sample efficiency.**  Improving sample efficiency is a central goal in off-policy RL. REDQ (Chen et al., 2021) addresses this by using large ensembles and high UTD ratios. However, REDQ relies on a fixed aggregation strategy and does not adapt to evolving uncertainty or training dynamics. AQE (Wu et al., 2022) and TQC (Kuznetsov et al., 2020) enhance learning by either averaging over subsets of multi-head critics (AQE) or truncating the highest $Q$-value estimates (TQC). However, both approaches require manually chosen thresholds or hyperparameters that must be tuned separately for each task or training regime. In contrast, DEA dynamically adapts its aggregation behavior based on ensemble disagreement, automatically adjusting conservatism and exploration without relying on fixed schedules or per-task tuning.

Other methods like SUNRISE (Lee et al., 2021) and MeanQ (Liang et al., 2022) also promote sample-efficient learning but are tailored to fixed schedules and architectures. DEA instead modulates ensemble aggregation dynamically, allowing a single actor-critic algorithm to perform robustly across both interactive (low-UTD) and sample-efficient (high-UTD) regimes without manual tuning.

DEA aims to learn a general-purpose aggregation mechanism that operates effectively across learning regimes, from interactive (low-UTD) to sample-efficient (high-UTD) settings, by leveraging ensemble disagreement in a principled, learnable manner. In this landscape, SAC (Haarnoja et al., 2018a) and REDQ (Chen et al., 2021) represent the most adopted baselines for each learning regime.

## 9 DISCUSSION

DEA is an adaptive ensemble-based method for actor-critics that works across learning regimes. Its key strength is adaptability: DEA learns directional aggregation weights that evolve with the data, allowing it to adjust autonomously to varying tasks, uncertainty levels, and training dynamics. This flexibility allows DEA to perform reliably across both interactive and sample-efficient settings, something that prior methods typically address only in isolation.

**Limitations.**  Despite strong empirical performance, DEA has limitations. Learnable aggregation parameters increase training complexity. Although DEA was stable across tasks and seeds in our experiments, its effectiveness may decline with very small ensembles that provide weak disagreement signals. Our study focused on dense-reward environments where random exploration often suffices and estimation bias can be partly offset by heuristics. The fact that DEA still yields clear gains highlights its strength, and we expect these benefits to be even greater in sparse-reward settings where exploration is harder. While results indicate that directional aggregation mitigates overestimation bias and improves learning, this remains unsupported by theory.

**Future work.**  Our work focuses exclusively on online RL; extending these ideas to the offline setting is an exciting direction, especially building on the promising findings demonstrated in the online setting. Additionally, pairing DEA with more advanced frameworks, such as BRO (Nauman et al., 2024b) or Simba (Lee et al., 2025), could further improve learning.

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
