# OpenReview forum: "Directional Ensemble Aggregation for Actor-Critics"
_ICLR.cc/2026/Conference — ICLR 2026 Conference Withdrawn Submission_

### Official Review · Reviewer_zbpB · 2025-10-23

**Soundness:** 2
**Presentation:** 3
**Contribution:** 2
**Rating:** 4
**Confidence:** 3

**Summary:**

The paper introduces Directional Ensemble Aggregation (DEA), a novel method designed to improve Q-value estimation in off-policy reinforcement learning (RL). Traditional actor-critic methods  often rely on static aggregation strategies, such as taking the minimum Q-value from an ensemble of critics, which can lead to  loss of valuable information. DEA aims to replace these static rules with a dynamic, learnable approach that adjusts based on training dynamics.

**Strengths:**

1. DEA presents a unique, learnable aggregation method that can learn from  task-specific uncertainties and ensemble dynamics. This is a significant advancement over traditional methods that use fixed aggregation strategies, which can hinder performance and adaptability.
2. The paper provides empirical evidence demonstrating DEA's effectiveness across various benchmarks, outperforming existing methods like SAC and REDQ.
3. The use of metrics such as Final Return, InterQuartile Mean (IQM), and Area Under the Learning Curve (AULC) provides a comprehensive evaluation of the method's effectiveness

**Weaknesses:**

1. The paper lacks a  theoretical analysis to prove the convergence of new TD update. Specifically, as shown in Equation (4), an extra term $\bar{k}\cdot \bar{\delta}(s,a)$ is introduced for TD update. This may affect the convergence of the operator.
  2. The introduction of learnable parameters for aggregation increases the complexity of the algorithm. This complexity may require more careful tuning and could pose challenges in terms of computational efficiency and stability during training.
  3. In ensemble disagreement, $\delta$ is calculated by averaging the differences between the valuations of any two Q-nets in the Q-net set. The computational complexity of is $O(N^2)$. Why not choose the standard deviation of multiple Q-net estimation?
  4. Why choose to train two k and two $\delta$?

**Questions:**

1. Why are the results in Figure 3 and Table 2 inconsistent?
 2. Why choose to train two k and two $\delta$? Are the learnable parameters redundant?
 3. Is the current delta calculation method reasonable?

---

### Official Review · Reviewer_h1ne · 2025-10-27

**Soundness:** 2
**Presentation:** 2
**Contribution:** 2
**Rating:** 4
**Confidence:** 3

**Summary:**

This paper proposes Directional Ensemble Aggregation (DEA), a fully learnable aggregation method that replaces static aggregation with a dynamic mechanism, allowing interpolation between conservative and explorative strategies as training progresses.

**Strengths:**

This paper proposes Directional Ensemble Aggregation for DRL and conducts extensive experiments across various tasks.

**Weaknesses:**

1. Can the definition of Ensemble disagreement vary? How would different definitions impact performance?
2. How does Ensemble disagreement and the performance of the proposed method change as the number of critics increases? It is recommended to analyze this through experiments.
3. It is suggested that the authors provide theoretical proof to demonstrate: 1) the convergence of the proposed method, and 2) why the proposed method outperforms SOTA solutions.
4. The baseline comparisons are too few; at least five more SOTA solutions should be included for comparison.
5. Table 3 should also present the standard deviation from 10 seed runs to show the variability of each method.
6. Intermediate results should be presented to demonstrate how the proposed method improves exploration capabilities (e.g., by showing policy entropy) or Q-value estimation (the difference from true Q-values) compared to SOTA solutions.

**Questions:**

1. Can the definition of Ensemble disagreement vary? How would different definitions impact performance?
2. How does Ensemble disagreement and the performance of the proposed method change as the number of critics increases? It is recommended to analyze this through experiments.
3. It is suggested that the authors provide theoretical proof to demonstrate: 1) the convergence of the proposed method, and 2) why the proposed method outperforms SOTA solutions.
4. The baseline comparisons are too few; at least five more SOTA solutions should be included for comparison.
5. Table 3 should also present the standard deviation from 10 seed runs to show the variability of each method.
6. Intermediate results should be presented to demonstrate how the proposed method improves exploration capabilities (e.g., by showing policy entropy) or Q-value estimation (the difference from true Q-values) compared to SOTA solutions.

---

### Official Review · Reviewer_nfo9 · 2025-10-29

**Soundness:** 2
**Presentation:** 3
**Contribution:** 2
**Rating:** 2
**Confidence:** 4

**Summary:**

The paper introduces Directional Ensemble Aggregation (DEA) for off-policy actor–critic RL. Instead of using a fixed rule to combine critic heads, DEA learns two small scalar parameters to build the target critic used for Bellman updates and the actor-update critic used for policy improvement respectively. Both parameters are adapted online from the ensemble’s internal disagreement signal, and their learning uses only the direction of the prediction error (over/under-shoot) to keep updates stable and noise-tolerant. Conceptually, DEA decouples how conservative the target should be from how optimistic the actor update can be, allowing the method to interpolate automatically between cautious and exploratory behavior as uncertainty changes during training. The approach integrates into a standard SAC pipeline with minimal code changes and no architectural overhead. Experiments on MuJoCo and DeepMind Control report improvements over SAC and REDQ when results are aggregated across the interactive and sample-efficient regimes.

**Strengths:**

1. Simple and well-grounded intuition: The method introduces a clean, decoupled aggregation mechanism for the target and actor critics that integrates seamlessly into SAC/REDQ pipelines with minimal engineering effort. Using ensemble disagreement to adjust optimism and conservatism is conceptually sound, and the sign-only adaptation is a practical way to stabilize updates and reduce variance.
2. Clear and interpretable: The paper is easy to follow, and the explanations of how the two learned parameters evolve during training help readers understand the method’s behavior in practice.

**Weaknesses:**

1. Unfair aggregation across regimes: The paper reports results by averaging performance across the interactive and sample-efficient regimes, but these two settings differ substantially in both algorithmic design intent and compute budgets (UTD ratio, ensemble size, and wall-clock cost). SAC is primarily designed for the interactive regime, emphasizing stability and low update-to-data ratios, while REDQ is optimized for sample efficiency with higher update ratios and larger ensembles. Aggregating their results into a single metric mixes incompatible evaluation conditions and obscures each method’s actual strengths. As a result, the reported average performance does not reflect a fair or meaningful comparison—DEA’s improvement could stem from outperforming SAC in one regime and underperforming against REDQ in the other, yet the aggregation hides this distinction. A clear, per-regime comparison is needed to judge where DEA truly provides benefits.
2. Insufficient ablations despite the claim: The paper states that spanning two regimes “naturally” serves as ablations over ensemble size and UTD. In practice, only the aggregated performance of two points is shown (ensemble size 2 vs. 10 with paired UTD settings). This does not characterize sensitivity or robustness
3. Unclear magnitude of gains: Even in the aggregated results, DEA’s improvement is not clearly demonstrated. While the method shows noticeable gains in area-under-the-learning-curve (AULC), the improvements on final performance and IQM are marginal. The paper should report the mean and median percentage improvements as well as the standard deviations across runs to better illustrate the actual magnitude of performance gains.

**Questions:**

See weaknesses

---

### Official Review · Reviewer_oa1b · 2025-10-29

**Soundness:** 2
**Presentation:** 3
**Contribution:** 2
**Rating:** 2
**Confidence:** 5

**Summary:**

This paper addresses reliable Q-value estimation in off-policy reinforcement learning for continuous control and highlights limitations of standard actor-critic methods: they mitigate overestimation bias by conservatively aggregating Q-value ensembles (e.g., taking the minimum), which discards useful information, cannot adapt to training dynamics, and generalizes poorly across learning regimes. To address these issues, this paper proposes Directional Ensemble Aggregation (DEA), a fully learnable aggregation method that replaces static rules with a dynamic mechanism capable of interpolating between conservative and explorative strategies as training progresses. DEA uses a decoupled aggregation with two learnable directional parameters: $\(\bar{\kappa}\)$, which constructs the aggregation that guides critic learning, and $\(\kappa\)$, which constructs the aggregation that guides actor learning. Both parameters are learned directly from data using disagreement-weighted Bellman errors, with updates depending only on the sign of the error for each sample. This design allows DEA to adjust automatically to task-specific uncertainty, ensemble size, and update frequency.

Empirically, across MuJoCo and the DeepMind Control Suite, DEA maintains reliable learning dynamics and outperforms static ensemble aggregation methods in both interactive and sample-efficient learning regimes, demonstrating strong generalization across settings where static rules often fail.

**Strengths:**

1) The paper is clearly written and provides a well-structured explanation of the motivation behind DEA and its design, making the proposed dynamic aggregation mechanism easy to follow and understand.
2) The paper conducts extensive experiments on two benchmark suites, MuJoCo and the DeepMind Control Suite, and the results demonstrate that DEA consistently outperforms both interactive methods (SAC) and sample-efficient methods (REDQ).

**Weaknesses:**

1) The paper lacks theoretical analysis or convergence guarantees to support the proposed method, making its stability and effectiveness rely solely on empirical evidence.
2) The proposed method introduces high computational complexity, especially due to the need to compute ensemble disagreement, yet the paper does not provide any analysis or discussion of the associated computational or resource overhead.

**Questions:**

1) Figures 10-13 show that the ensemble disagreement decreases significantly as training progresses. However, in Figure 1, the two directional parameters $\kappa$ and $\bar{\kappa}$ do not exhibit a clear trend of change, especially after 0.2M interactions. Intuitively, as the ensemble disagreement decreases over training, $\kappa$ is expected to gradually increase to promote more optimistic exploration. Could the authors clarify why this expected pattern is not observed?
2) At the beginning of training, $\kappa$ is initialized to zero while $\bar{\kappa}$ is initialized to a negative value. Could the authors explain the motivation behind these choices? A brief justification would help readers better understand the initialization strategy and the intended behavior of DEA during early training.
3) Furthermore, ensemble disagreement is typically large during the early phase of training. In this case, one might expect $\bar{\kappa}$ to be small or even negative to avoid overestimation. However, the results show that $\bar{\kappa}$ is relatively large at the beginning compared to its later values. Could the authors elaborate on why this occurs?

---

### Note · Authors · 2025-11-14

I have read and agree with the venue's withdrawal policy on behalf of myself and my co-authors.